# Typical thermalization of low-entanglement states
Christian Bertoni [1] ✉, Clara Wassner[1], Giacomo Guarnieri [1,2] & Jens Eisert [1,3] ✉

Proving thermalization from the unitary evolution of closed quantum systems is one of the oldest questions that is still only partially resolved. Efforts led to various versions of the eigenstate thermalization hypothesis (ETH), which implies thermalization under certain conditions. Whether the ETH holds in specific systems is however difficult to verify from the microscopic description of the system. In this work, we focus on thermalization under local Hamiltonians of low-entanglement initial states, which are operationally accessible in many natural physical settings, including schemes for testing thermalization in experiments and quantum simulators. We prove thermalization of these states under precise conditions that have operational significance. More specifically, motivated by arguments of unavoidable finite resolution, we define a random energy smoothing on local Hamiltonians that leads to local thermalization when the initial state has low entanglement. Finally we show that this transformation affects neither the Gibbs state locally nor, under generic smoothness conditions on the spectrum, the short-time dynamics.

Not long after their formulation, it became clear that the postulates of quantum mechanics were, to a degree at, odds with the principles of statistical mechanics[1]. On the one hand, closed quantum systems evolve unitarily, thus preserving information about their initial conditions. On the other hand, statistical mechanics is centered around concepts such as irreversibility and thermalization, which allow to describe many-body systems in terms of just a few macroscopic quantities and parameters that depend neither on the initial conditions nor on the microscopic details[2–6]. Both theories have nowadays provided correct predictions of uncountable experimental observations. Thermalization of closed many-body quantum systems is in particular observed to occur in practice with overwhelming evidence, also in quantum simulators which allow to probe their dynamics with high levels of precision[7–12]. This apparent contradiction is eased once one only requires the expectation value of a subset of "physical" observables, for example, local observables, to agree with the predictions of statistical mechanics. As a matter of fact, in a lattice system, the state of the system reduced to local patches can relax to a thermal state while the global state evolves unitarily and remains pure. Nevertheless, a derivation of thermalization in this sense from the microscopic description of the dynamics has remained largely elusive. Efforts in this direction have led to various formulations of the eigenstate thermalization hypothesis (ETH)[13,14], which posits that each eigenstate of a thermalizing Hamiltonian provides the same expectation values of physical observables as the ones given by local thermal states. The ETH, however, is difficult to verify starting from a microscopic model. In addition, the precise conditions on the initial state and on the

observables leading to thermalization are highly sensitive to the formulation of the hypothesis. Moreover, the absence of thermalization, as observed in integrable or localized systems[15–17], seems to require particular, carefully set-up conditions. This has led to the idea of typicality, i.e., that "most" physical systems obey statistical mechanics. This usually means that if the system (for instance, it's Hamiltonian) is drawn at random from a reasonable set, with overwhelming probability, thermalization occurs as expected, and while any given system under examination is usually not randomly drawn, it should behave like a typical one unless explicitly engineered not to. Works in this direction[18–22] have contributed to formalizing the idea that thermalization is the rule, rather than the exception, even in isolated systems.

A make-or-break question is left open, however: why would one generically expect—starting with natural initial states such as low-entanglement states—quantum systems to relax to states that are operationally indistinguishable from Gibbs states by local measurements? This is the actual empirical observation of apparent thermalization that needs to be answered by a theoretical framework. The core prejudice to be overcome is not to prove thermalization for all Hamiltonians but only for most such Hamiltonians that are slightly randomized and still yield extremely similar dynamical predictions compared to the original Hamiltonian.

In this work, we prove that initial low-entanglement states thermalize under appropriately weakly randomized local Hamiltonians. Thermalization under randomized or typical Hamiltonian has been considered and proven in previous works, where the randomization can either be in the form of an additive random matrix[23–25] or, like in this work, a random

[1]Dahlem Center for Complex Quantum Systems, Freie Universität, Berlin, Germany. [2]Department of Physics, University of Pavia, Pavia, Italy. [3]Helmholtz Center Berlin, Berlin, Germany. ✉e-mail: chr.bertoni@gmail.com; jense@gmail.com

rotation of the eigenbasis[20,21]. In contrast to these works, we focus on making the randomization as weak as possible while still ensuring thermalization under the randomized Hamiltonian. Under a meaningful spectral smoothness assumption, the randomization can be chosen to be weak enough such that it does not affect the short-time dynamics. In addition, we show that the randomization does not affect the thermal properties of the Hamiltonian. With these last two points, we take steps towards bridging the gap between the spatially local description of the system given by a local Hamiltonian, and the energy-local description given by random matrix theory necessary to explain thermalization. Accepting that the Hamiltonian will anyway only be known to finite precision, such a perturbative typicality argument contributes to explaining why a dynamical convergence to Gibbs states is so ubiquitous in nature.

## Methods

It is known that quantum lattice models that are prepared in typical states that are restricted to a narrow window centered around a given energy—invoking notions of equivalence of ensembles—will give rise to local expectation values that are indistinguishable from thermal ones[26]. While this is conceptually insightful, it is not clear how such states can arise from the physical dynamics of natural initial states. What is, in contrast, very natural is to consider low-entanglement initial conditions and natural time evolution. Such low-entanglement states, in addition to being fundamental to the study of lattice systems, are commonly the only accessible initial states to thermalization experiments in quantum simulators or numerical examination of thermalization, for example, in the context of probing many-body localization[27–31], and are often considered in works on relaxation and equilibration[32–34].

Concretely, given a local Hamiltonian $H$ and an initial low-entanglement state $\rho$, we define an ensemble of random Hamiltonians such that, if $H'$ is a Hamiltonian drawn from this ensemble: 1. the Gibbs states of $H$ and $H'$ are locally indistinguishable, and 2. with overwhelming probability the equilibrium state resulting from the unitary evolution $e^{-iH't}\rho e^{iH't}$ is locally indistinguishable from said Gibbs state. Importantly, we further give an assumption under which the dynamics of $H$ and $H'$ are indistinguishable in the short-time regime. (See Fig. 1).

We address this by studying states in an analog of the micro-canonical ensemble, which is allowed to have support on the whole energy spectrum. Building upon and extending the work of ref. 26, we show that, under precise and physically meaningful conditions, these states are locally equivalent to Gibbs states. We later show that such states arise from the long-time dynamics of low-entanglement initial states under typical Hamiltonians. Our typicality transformation, defining the ensemble of random Hamiltonians, consists of a unitary operation that randomly mixes eigenstates with

nearby energies. Related transformations have been introduced in ref. 35 to compute higher-order correlation functions under a generalized version of the ETH stemming from local (in energy) unitary invariance, and this has been connected to the theory of free probability in ref. 36. In our work, we explicitly demonstrate how such a transformation leads to equilibrium states being locally indistinguishable from thermal states. Moreover, by elucidating how the original Hamiltonian is related to the transformed one, we motivate a meaningful, operational interpretation of our Hamiltonian-typicality approach: Given a physical system in the right conditions, one can find a local Hamiltonian $H$, which well describes the dynamics. In realistic situations, it is difficult to show whether $H$ thermalizes or not. Nevertheless, there is a host of other models $H'$ are close to $H$, lead to thermalization and, if $H$ is well-behaved, describe very similar dynamics as well as equilibrium properties. $H$ cannot be known to infinite precision, and this operational uncertainty can be modeled by our Hamiltonian ensemble.

Throughout this work, we consider a cubic $D$-dimensional lattice $\Lambda$ with $N$ sites. With each site $i$ we associate a Hilbert space $\mathcal{H}_i$ of dimension $d$. $H$ is a $k$-local Hamiltonian on $\Lambda$ for a constant $k$, meaning that it is of the form $H = \sum_{i \in \Lambda} h_i$ where $h_i$ has operator norm 1 and is only supported on sites $j$ s.t. $d(i,j) \le k$. Here, $d(\cdot, \cdot)$ is the standard Manhattan distance on the lattice. We will denote the spectral decomposition of $H$ as $H = \sum_\nu E_\nu P_\nu$. For ease of presentation, in the main text, we will assume the eigenstates to be non-degenerate, i.e., $\mathrm{tr}(P_\nu) = 1$ for all $E_\nu$. We will use standard $O$ and $\Omega$ notation, and we will use $\tilde{O}, \tilde{\Omega}$ to denote asymptotic upper bounds where logarithmic factors are ignored, i.e., $f(h) = \tilde{O}(g(h))$ iff $f(h) = O(\log^r(h)g(h))$ for some $r$. Before discussing thermalization, we need to discuss a natural prerequisite: equilibration[6,37–39]. Intuitively, a state equilibrates if, after a finite amount of time, the system reaches an invariant state. This means that if one looks at the whole evolution as time tends to infinity, the system will spend most of its time close to equilibrium, and hence the state at equilibrium should agree with the infinite-time averaged state

$$\rho_\infty^H = \lim_{T \to \infty} \frac{1}{T} \int_0^T dt\, \rho^H(t), \tag{1}$$

with $\rho^H(t) = e^{iHt}\rho e^{-iHt}$. The system is then said to equilibrate if the fluctuations around the average for an observable $A$ are small in the sense of

$$\Delta A_\infty := \lim_{T \to \infty} \frac{1}{T} \int_0^T dt\, \mathrm{tr}\big(A(\rho^H(t) - \rho_\infty^H)\big)^2 \xrightarrow{N \to \infty} 0. \tag{2}$$

Equilibration, in this sense, has been rigorously proven in a variety of settings[39–41]. Thermalization essentially consists of a stronger requirement of equilibration, where the equilibrium state must coincide with the thermal (Gibbs) state

$$g_\beta(H) = \frac{e^{-\beta H}}{Z}, \quad Z = \mathrm{tr}(e^{-\beta H}). \tag{3}$$

In particular, we will be interested in understanding how distinguishable a given state is from a Gibbs state, given only access to local observables. We introduce the quantity

$$D_l(\rho, \sigma) := \frac{1}{|\mathcal{C}_l|} \sum_{C \in \mathcal{C}_l} \| \rho_C - \sigma_C \|_1, \tag{4}$$

where $\mathcal{C}_l$ denotes the set of all hypercubes in the lattice of side length $l$ and $\rho_C$ is the state $\rho$ reduced to $C$. If $\rho, \sigma$ are translationally invariant, this is simply equal to the trace norm $\| \rho_C - \sigma_C \|_1$ for any $C \in \mathcal{C}_l$. Otherwise, it measures the distinguishability of $\rho$ and $\sigma$ given access to observables of the form $\frac{1}{|\mathcal{C}_l|} \sum_{C \in \mathcal{C}_l} O_C$, such as typical average local quantities in statistical physics. We will then say that a state is locally thermal if $D_l(\rho, g_\beta(H))$ converges to 0 with $N$ for some $\beta$. In the remainder of this work, we will make some

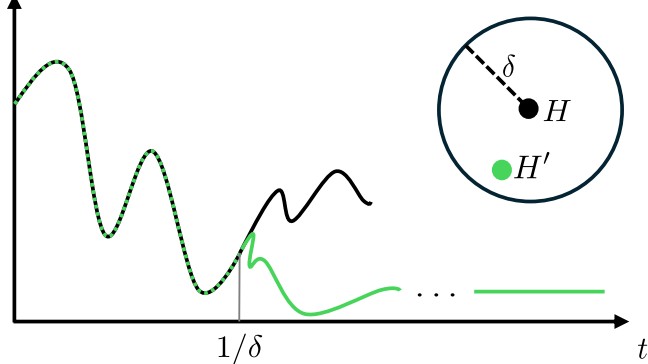

**Fig. 1 | Illustration of the dynamics of the original and perturbed Hamiltonians.** The distance of the new Hamiltonian from the original one is bounded by a parameter $\delta$. The plot shows how the evolution of the expectation value of an observable under $H$ or $H'$ behaves, where $H$ is the original Hamiltonian and $H'$ the randomized one: the two evolutions agree up to a time $\sim 1/\delta$, and the perturbed Hamiltonian equilibrates to a thermal value.

assumptions on the Gibbs state of the system under examination, $g_\beta(H)$. First, we will require it to have exponential decay of correlations. A state $\rho$ is said to have exponential decay of correlations if for some $\xi > 0$, for any two regions $X, Y \subset \Lambda$ and $A, B$ supported on $X$ and $Y$, respectively,

$$|\langle AB \rangle_\rho - \langle A \rangle_\rho \langle B \rangle_\rho| \le \parallel A \parallel \parallel B \parallel e^{-d(X,Y)/\xi}. \tag{5}$$

The Gibbs state $g_\beta(H)$ always has an exponential decay of correlations in one dimension[42], as well as in any higher dimension above a certain critical temperature[43] that depends only on a few parameters of the system. We will denote by $\sigma$ the standard deviation of the energy, i.e., $\sigma^2 = \mathrm{tr}(g_\beta(H)H^2) - \mathrm{tr}(g_\beta(H)H)^2$, and assume $\sigma^2 \ge \Omega(N)$, which implies that the specific heat capacity is non-zero in the thermodynamic limit.

## Results

The micro-canonical ensemble is commonly defined as the maximally mixed state-supported in a narrow window around a fixed energy. This physically models maximal uncertainty with an energy constraint. However, arbitrary and even physically motivated states, such as low-entanglement states, can have non-trivial support over the whole spectrum during the whole time evolution; these states will, therefore, never be micro-canonical in the sense above, particularly not at equilibrium. With the goal of overcoming this issue, we leverage techniques used to prove equivalence between micro-canonical and canonical ensembles. Such equivalence results have a long history, with the first proofs for lattice systems in the strict thermodynamic limit being given by Lima[44,45]; later, Brandão and Cramer[26] generalized this to finite-sized lattice systems. In this latter work, it is proven that equivalence with a canonical ensemble is already achieved by states confined in a micro-canonical window that are not maximally mixed but have sufficiently high entropy and an expected energy that is sufficiently close (but not necessarily equal to) the expected energy of the Gibbs state. We adapt this result to the situation in which the state is not confined to a window but instead has support over many such windows, plus decaying tails. First of all, we call this a generalized micro-canonical ensemble, meant to capture the thermal behavior of states that are supported on regions of the spectrum larger than what the usual micro-canonical ensemble allows.

**Definition 1**. (Generalized micro-canonical ensemble (GmE)) Let $[E - \Delta, E + \Delta]$ ($\Delta > 0$) denote an energy window centered around a value $E$ and divided into $K$ bins of various size $\delta_k$, with $k = 1, \ldots, K$. Let $\delta = (\delta_1, \ldots, \delta_K)$; we define a generalized micro-canonical ensemble (GmE) to be the state of the form

$$\omega := \omega(E, \Delta, \delta, \mathbf{q}) = \sum_{k=1}^{K} q_k \omega_{\delta_k} \tag{6}$$

where $\omega_{\delta_k}$ is the micro-canonical ensemble supported inside the window $k$, and where $\mathbf{q} = (q_1, \ldots, q_K)$ such that $\sum_{k=1}^{K} q_k = 1$.

This state, therefore, physically represents a statistical combination of micro-canonical ensembles; see Fig. 2. For the sake of simplicity, we will choose $\delta_k = \delta$ from now on. However, all results are shown in the Supplementary Information with this assumption relaxed, unless otherwise specified. Before stating our first main result, we need to introduce the notion of the *Berry-Esseen* (BE) error, which quantifies the difference between a state written in the energy eigenbasis and a Gaussian distribution. More specifically, if $\Pi_x$ is the projector onto all energy eigenstates with energy smaller than $x$, then the BE error of $\rho$ with respect to $H$ is defined as $\zeta_N = \sup_x |\mathrm{tr}(\rho\Pi_x) - G(x)|$, where $G(x)$ is the Gaussian distribution with the same mean and variance as $\rho$. It was proven that if $\rho$ has exponential decay of correlations, then $\zeta_N \le \tilde{O}(N^{-1/2})$[26]. Simple examples of saturating this bound are known, and this bound is expected to be saturated by certain non-thermalizing models. Nonetheless, under some more generic constraints, such as highly entangled eigenstates, a more favorable scaling is

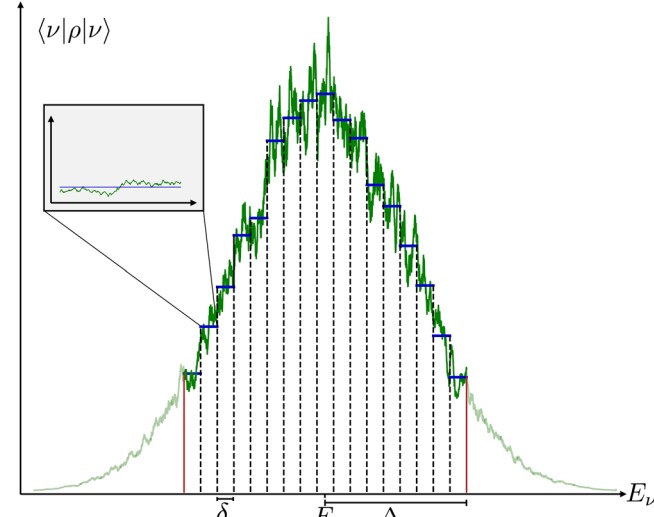

**Fig. 2 | Cartoon illustration of the setting of Definition 1 and Definition 2.** In this example, the state is diagonal in the energy eigenbasis, and it is represented as a probability distribution. The blue line is a Generalized microcanonical Ensemble (GmE) state, and the green line an approximate GmE state. The insert shows both states restricted to one of the windows.

expected, even up to $\zeta_N \le e^{-\Omega(N)}$ for product states[46]. From now on, we denote by $\zeta_N$ the BE error with $\rho = g_\beta(H)$. In what follows, we will assume $\zeta_N \le \tilde{O}(N^{-1/2-\kappa})$ for some $\kappa \ge 0$. This includes the worst case $\kappa = 0$.

**Theorem 1**. (Ensemble equivalence) Let $H$ be a local Hamiltonian and $\beta$ an inverse temperature for which the Gibbs state $g_\beta(H)$ has exponential decay of correlations, standard deviation $\sigma \ge \Omega(\sqrt{N})$ and Berry-Esseen error $\zeta_N \le \tilde{O}(N^{-1/2-\kappa})$ for $\kappa \ge 0$. Let $\omega$ denote a GmE with $\Delta, \delta$ satisfying

$$e^{\Delta^2/\sigma^2} \le \tilde{O}\left(N^{\frac{1-\alpha}{D+1}}\right), \quad \Omega\left(N^{\frac{1-\alpha}{D+1}-\kappa}\right) \le \delta \le \sigma, \tag{7}$$

with $\alpha \in [0, 1)$ and such that $|E - E_\beta| \le \sigma$. Then for any side length $l$ such that $l^D \le C_1 N^{\frac{1}{D+1}-\gamma_1\alpha}$, the following holds

$$D_l(\omega, g_\beta(H)) \le C_2 N^{-\gamma_2\alpha}, \tag{8}$$

with $C_1, C_2$ being system-size independent constants, and $\gamma_1, \gamma_2$ only depend on the dimension of the lattice $D$.

This first main result shows that, for appropriate choices $\Delta$ and $\delta$, GmE states are locally indistinguishable from Gibbs states. A GmE state can be seen as a mixture of micro-canonical states spanning a range of temperatures and Theorem 1 shows that as long as its range is small enough, the state still looks thermal with a well-defined temperature. Notice that if $\kappa > 0$, i.e., if the BE error is better than the worst case scenario, $\delta$ can be chosen to decay with the system size. This window size will play a crucial role in determining the minimal "strength" of the randomization necessary to enforce thermalization; our bounds on $\delta$ are inherited from the techniques of ref. 26. It is worth noting that later works significantly improved this window size for the equivalence of the canonical and micro-canonical ensemble, up to sizes $\delta \sim e^{-\sqrt{N}}$[47,48], but these results are either dependent on an exactly micro-canonical state (with maximum entropy) or on the micro-canonical state having an exactly defined energy, which makes their application for our purposes difficult.

Keeping in mind our initial goal of capturing equilibrium states resulting from natural and physically motivated initial states, it may seem artificial to consider only block-like states with sharp jumps between energy intervals. Therefore, postponing the discussion about their physicality to the next Section below, we, first of all, prove that the same ensemble equivalence

holds if the state's structure gets more relaxed, i.e., if it is only approximately GmE in a sense precisely elucidated below.

**Definition 2**. (Approximate GmE) Let $E$, $\Delta$, $\delta$, **q** be as in Definition 1. We define $\omega_\eta$ an approximate GmE if it is of the form

$$\omega_\eta = p_\Delta \left( \sum_{k=1}^{K} q_k \tilde{\omega}_{\delta_k} \right) + (1 - p_\Delta)\rho_{\text{tail}} \tag{9}$$

and its von Neumann entropy satisfies

$$\sum_{k=1}^{K} q_k (S(\omega_{\delta_k}) - S(\tilde{\omega}_{\delta_k})) \leq \eta, \tag{10}$$

with $\tilde{\omega}_{\delta_k}$ being defined on the Hilbert space spanned by the eigenstates in the $k$-th energy bin, and $\rho_{\text{tail}}$ on the Hilbert space spanned by the eigenstates outside $[E - \Delta, E + \Delta]$.

This state represents a more physical version of a GmE state inside the energy window $\Delta$, with decaying tails outside, that has an entropy $\eta$-close to the maximum one. Importantly, in Supplementary Note 1, we demonstrate that Theorem 1 holds true also for the approximate GmE and takes the form

$$D_l(\omega_\eta, g_\beta(H)) \leq C_2 N^{-\gamma_2 \alpha} + 2(1 - p_\Delta), \tag{11}$$

with $\eta \leq N^{\frac{1-\alpha}{D+1}}$.

This shows that states that are concentrated around an energy regime and are sufficiently "smooth" are locally equivalent to Gibbs states. The results above are a generalization of the equivalence of ensembles result of ref. 26, and their proof is presented in Supplementary Note 1.

The next question is whether (approximate) GmE states can actually be obtained from the Hamiltonian evolution of isolated systems under natural or typical conditions. There are two main aspects we consider when talking about "natural" conditions: (i) the Hamiltonian responsible for the time evolution and (ii) the initial state of the system. Regarding (i), we consider typical Hamiltonians in the sense that we will make precise below in order to exclude edge cases or fine-tuned Hamiltonians for which one does not expect thermalization (for instance, integrable models). Concerning (ii), previous typicality approaches have assumed the initial state to be confined in a well-defined energy interval, and have shown properties of the relaxation towards a micro-canonical ensemble in said interval. Here, instead, we start from the assumption of exponential decay of correlations, i.e., low entanglement between spatially separated regions, which we take as natural starting states for lattice systems. These states have been shown to have fast decaying tails in energy[49] which makes them ideal candidates to flow to approximately GmE states. Here, for simplicity of presentation, we focus on the case of product states and leave the more general case of states with exponential decay of correlations and the proofs to Supplementary Note 3.

Let us expand on the ensemble of typical Hamiltonians that we consider. Starting from any local Hamiltonian on the lattice, we divide its energy spectrum into energy intervals of equal width $\delta$ which we call $I_k$, for $k = 1, \cdots, K$. The eigenstates contained within each interval span a vector space which we call $\mathcal{W}_k$. We then consider unitaries of the form $U = \bigoplus_k U_k$, where $U_k$ is drawn from the Haar measure of the unitary group acting on $\mathcal{W}_k$. This defines an ensemble of random unitaries which we denote as $\mathcal{E}(\delta)$. A typical Hamiltonian is then $UHU^\dagger$ for such a random unitary $U$. All these Hamiltonians have the same spectrum as the original local Hamiltonian $H$, and the randomization given by $U$ is designed to preserve the expected energy of any state.

The following is a consequence of measuring concentration and the results of ref. 49 about the energy tails of product states.

**Lemma 1**. (Approximate GmE at equilibrium) Let $\rho$ be a product state and $H$ be a local Hamiltonian. Let $\rho_\infty^{UHU^\dagger}$ be defined as in Eq. (1), where $U$ is drawn from $\mathcal{E}(\delta)$. Consider the interval $I = [E - \Delta, E + \Delta]$ around $E = \text{tr}(\rho UHU^\dagger)$ with $\Delta \geq \omega(\sqrt{N})$ an integer multiple of $\delta$, then

$$\rho_\infty^{UHU^\dagger} = p_\Delta \left( \sum_{k:I_k \subset I} q_k \tilde{\omega}_{\delta_k} \right) + (1 - p_\Delta)\rho_{\text{tail}} \tag{12}$$

with $p_\Delta \geq 1 - e^{-c_1 \frac{\Delta^2}{N}}$, and for $r > 0$, with probability at least $1 - 2^{-r+1}$, we have

$$\sum_{k:I_k \subset I} q_k (S(\omega_{\delta_k}) - S(\tilde{\omega}_{\delta_k})) \leq r, \tag{13}$$

where $c_1$ is a system-size independent constant.

We have then the following consequence on typical thermalization.

**Theorem 2**. (Typical thermalization) Let $H$ be a $k$-local Hamiltonian and $\rho$ be a product state. Let $g_\beta(H)$ be the Gibbs state of $H$ at inverse temperature $\beta$ such that $|\text{tr}(g_\beta(H)H) - \text{tr}(\rho H)| \leq \sigma$. Assume $g_\beta(H)$ has exponential decay of correlations, $\sigma \geq \Omega(\sqrt{N})$, and $\zeta_N \leq \tilde{O}(N^{-1/2-\kappa})$. For any constant $\alpha \in [0, 1)$, choosing $\delta = \Omega(N^{\frac{1-\alpha}{D+1}-\kappa})$, with probability at least $1 - \exp(-c_2 N^{\frac{1-\alpha}{D+1}})$ drawing $U$ at random from $\mathcal{E}(\delta)$, we have

$$D_l(\rho_\infty^{UHU^\dagger}, g_\beta(H)) \leq C_2 N^{-\gamma_2 \alpha} + \tilde{O}(N^{-\gamma_3(1-\alpha)}), \tag{14}$$

where $c_2$, $C_2$, $\gamma_2$, $\gamma_3$ are system-size independent constants.

In Supplementary Note 3, we state and prove these results more generally for any state concentrated around its average, which includes states with exponentially decaying correlations. The consequence of this relaxation of the assumption is that the decay in the system size is quasi-polynomial rather than polynomial. We stress that we pick the specific case of low-entanglement states because their fast decaying tails allow for better approximation when the tails are cut outside of the window of size $\Delta$. As a matter of fact, our results hold for any state $\rho$, as long as its energy variance is linearly bounded $\sigma_\rho^2 = \text{tr}(\rho H^2) - \text{tr}(\rho H)^2 \leq O(N)$. We could then use Markov's inequality as opposed to the concentration bounds in[49] to show a linear decay of the tails in $\Delta^2/\sigma_\rho^2$. Since for Theorem 1 to apply, we must have $\Delta^2/\sigma_\rho^2 \sim \log(N)$, this leads only to bounds decaying very slowly as $\sim 1/\log(N)$ in Theorem 2. Theorem 2 shows that the equilibrium state is locally thermal; in the Supplementary Note 3 we show under mild spectral assumptions that the randomized Hamiltonian equilibrates with high probability to this state. Although it may seem strange at first glance that under the dynamics of $UHU^\dagger$ the state thermalizes to the Gibbs state of $H$ and not of $UHU^\dagger$, we prove that the Gibbs states of these two Hamiltonians are locally indistinguishable. More specifically, under the same assumptions as Theorem 2, for any $U$ drawn from $\mathcal{E}(\delta)$ we have

$$D_l(g_\beta(H), g_\beta(UHU^\dagger)) \leq O(N^{-\gamma_4 \alpha - \gamma_4 \kappa}) \tag{15}$$

for system-size independent constants $\gamma_4$, $\gamma_5$. The proof may be found in the Supplementary Note 3, and easily generalizes to other choices of $\delta$. On an intuitive level, this can be seen because both states are locally indistinguishable from a micro-canonical state, which is the maximally mixed state inside an energy window and, hence, invariant under rotations in the window. As anticipated, the unitary ensemble $\mathcal{E}(\delta)$ is chosen in order to approximately preserve the energy of any state; this implies that $U \sim \mathcal{E}(\delta)$ approximately commutes with the Hamiltonian, and we show $\|H - UHU^\dagger\|_\infty \leq \delta$. For the choice of $\delta$ as in Theorem 2, we derive the following result

$$\| e^{-iHt}\rho e^{iHt} - e^{-iH't}\rho e^{iH't} \|_1 \leq 2t\, O\left(N^{\frac{1-\alpha}{D+1}-\kappa}\right), \tag{16}$$

with $H' = UHU^\dagger$. This means that the dynamics under $H$ and $UHU^\dagger$ are indistinguishable up to a time $t^* \sim N^{-\frac{1-\alpha}{D+1}}$. If $\kappa > 0$, that is, the BE error decays faster than the worst-case scenario, $\alpha$ can be chosen such that $t^*$ increases with the system size. In other words, $H$ and $UHU^\dagger$ generate nearly the same dynamics for a time $t^* \sim \text{poly}(N)$. Finally, we would like to emphasize that our rigorous approach allows us to put on precise and solid ground some of the results obtained on equilibration in ref. 50.

It is now worth noting that if both $\rho$ and $\sigma$ are translation-invariant, the averaging over regions in the definition of $D_l(\rho, \sigma)$ can be dropped, making the indistinguishability statement valid for any observable supported on an individual small region. We show that if the original Hamiltonian is translation-invariant, we recover this property to some extent in the equilibrium state of the perturbed Hamiltonian. More specifically, consider an observable $A$ supported in $C \in \mathcal{C}_l$ and $H$ translation-invariant. For $U$ drawn from, $\mathcal{E}(\delta)$ we show, assuming all windows centered around an extensive energy contain exponentially many eigenvalues, that except with probability $e^{-\Omega(N)}$,

$$\left| \text{tr}\left( \left( g_\beta(H) - \rho_\infty^{UHU^\dagger} \right) A \right) \right| \leq$$
$$e^{-\Omega(N)} + D_l\left( \rho_\infty^{UHU^\dagger}, g_\beta(H) \right). \tag{17}$$

Details and proofs are available in the Supplementary Note 5. By applying this to $\rho = g_\beta(UHU^\dagger)$ we also get translation invariance in the same sense for the randomized Gibbs state, that is, $\rho_\infty^{UHU^\dagger}$ can be replaced by $g_\beta(UHU^\dagger)$ in Eq. (17).

Turning to notions of *dynamical thermalization*, we now investigate the typical time-evolution of the expectation value of a generic observable $A$, $\langle A \rangle_\rho := \text{tr}(A\rho)$, under the evolution generated by $UHU^\dagger$. In Supplementary Note 4, we show, assuming all windows centered around an extensive energy contain exponentially many eigenvalues, that except with probability $(N/\delta)^2 e^{-\Omega(N)}$, the time-evolution is bounded by

$$\left| \langle A \rangle_{\rho^{UHU^\dagger}(t)} - \langle A \rangle_{\rho_\infty^{UHU^\dagger}} \right| \leq e^{-\Omega(N)} + R(t), \tag{18}$$

where $R(t)$ is a function of $t$ depending on details of the spectrum of $H$, on $A$, and on $\rho$. Performing a similar analysis to the one in ref. 21 to our ensemble, and assuming that the spectrum in each window can be well approximated by a suitably flat continuous spectrum, we show that

$$R(t) \sim \| A \|_\infty \frac{N^2}{\delta^2 t^2}. \tag{19}$$

Hence, under this physical assumption, thermalization up to some $\epsilon$ is reached after a time $\sim N^{O(1)}/\epsilon$.

## Conclusions

In this work, we have made progress in the long-standing research quest of proving thermalization from first principles of quantum mechanics, by showing thermalization of low-entanglement states under typical Hamiltonians. In particular, we have defined ensembles that naturally emerge from the time-evolution of low-entanglement states under typical Hamiltonian evolution and show that they are locally indistinguishable from the Gibbs ensemble. The typicality in the Hamiltonian is given by randomizing the eigenbasis by scrambling eigenstates near in energy, and we show that this randomization does not affect the Gibbs state locally. Furthermore, we show that if the Gibbs state has a relatively generic smoothness property, the randomization does not affect the short-time dynamics in any discernible way.

On a higher level, this is one of the main physical messages of this work: it is unrealistic to assume that a Hamiltonian can be specified up to arbitrary precision. The ubiquity of thermalization may then be explained by considering that its absence requires instead a high precision in the specification of the system. In this light, proving thermalization for all or a large class of local Hamiltonians might not be required to explain this observation.

The technical results established here are also expected to be helpful in the design of quantum algorithms for preparing Gibbs states[51–53]. Relaxing the assumption of requiring the entire state to be globally indistinguishable from a Gibbs state (as suggested in ref. 53) may well be helpful in this endeavor.

Several natural open questions remain. Importantly, the problem of identifying precise properties of the Hamiltonian achieves a lower than worst-case BE error. For these Hamiltonians, our result predicts that a vast ensemble of other Hamiltonians exists that all have indistinguishable short-time dynamics and which eventually thermalize. Furthermore, while our randomization achieves thermalization, more work may be needed to explore the precise physical mechanisms that can be held responsible for them. On the one hand, being able to implement such transformations in a controlled way might have implications in the form of algorithms for Gibbs state preparation. On the other hand, understanding under what condition an uncertainty in either state or Hamiltonian preparation can translate into a random energy-preserving perturbation such as the one we defined might shed light on the stability of many-body localization under realistic conditions. It is the hope that this work contributes to the understanding that, under a "trembling hand", most systems follow the laws of quantum statistical mechanics.

## Data availability

Data sharing is not applicable to this article as no datasets were generated or analyzed during the current study.

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

## Acknowledgements

We thank Henrik Wilming, Ingo Roth, and Lennart Bittel for discussions during the early stages of this project. We also thank Álvaro Alhambra and Silvia Pappalardi for useful comments on the first draft. This work has been supported by the DFG (FOR 2724 and CRC 183), the BMBF (MUNIQCAtoms), the Quantum Flagship (PasQuans2), the FQXi, and the ERC (DebuQC).

## Author contributions

J.E. and C.B. initially formulated the project. C.B. and C.W. are responsible for the technical results. G.G. conceived the framing of the work in terms of ensemble equivalence and the definitions of GmE and aGmE. All authors partially contributed to all technical results and the writing of the manuscript.

## Funding

## Competing interests

The authors declare no competing interests.

## Inclusion and ethics

No ethical questions have been touched upon in this work.
