## [Transparent Peer Review file · Communications Physics]

Typical thermalization of low-entanglement states

Corresponding Author: Professor Jens Eisert

This manuscript has been previously reviewed at another journal that is not operating a transparent peer review scheme. The manuscript was considered suitable for publication with further review at Communications Physics.

Version 0:

Reviewer comments:

Reviewer #4

(Remarks to the Author)

The authors consider the problem of thermalization of systems described by local Hamiltonians. They argue that, since the true Hamiltonian cannot be known exactly, one can only assume a random “prior” distribution of Hamiltonians, preferably on a small set that is centered around the original Hamiltonian.

As their main result, the authors show that a slight randomization of the Hamiltonian can ensure the thermalization of initial states with exponentially decaying correlations.

The small set around the original Hamiltonian is defined via a smoothing parameter δ that limits the perturbation of the elements on the diagonal. The same δ sets the width of the belt around the diagonal within which it is allowed to perturb the nondiagonal elements.

Randomization of the Hamiltonian within these constraints is achieved by Haar-randomizing δ -sized blocks near the diagonal.

The authors show that the δ ensuring thermalization can be chosen to be much smaller than the average energy of the system, so much so that the perturbation barely affects the local states, the energy, and the short-time dynamics.

All previous results in the present context showed thermalization only when the full Hamiltonian is randomized, e.g., by Haar-randomly sampled rotations of the whole basis. Doing so perturbs the Hamiltonian so much that the energy fluctuates macroscopically, and the local states change drastically.

So, such randomization cannot be attributed to mere noise or lack of knowledge.

Thus, this work goes significantly beyond those works by proving thermalization with essentially undetectable randomization of the Hamiltonian that can be attributed to weak noise or lack of knowledge. All results are proved with a good level of mathematical rigor.

Together, the above points check all the boxes of the criteria for publication of Communications Physics. Therefore, the results in this paper more than merit being published in this journal.

There have already been three pretty thorough reviews of the paper, and the authors addressed most of the issues raised by the previous referees. As such, the revised paper is in good shape as is.

Below are some minor issues that the authors should fix before publication.

- Somewhere below Eq. (19), the authors write “randomizing the eigenbasis locally in energy”, which is a confusing use of language. The term local is already reserved for spatially locality, and the $H \rightarrow UHU^*$ transformation makes the Hamiltonian lose its local structure.

- Should the ω in the line above Eq. (12) be Ω ?

- In Eq. (17), the first ρ_∞ has a loose superscript.

Reviewer #5

(Remarks to the Author)

This paper analytically discusses the thermalization process in time evolution when the initial state has only a small amount of entanglement. There are several unique aspects in the analysis; for example, the use of a generalized micro-canonical ensemble (GmE).

The primary results are Theorem 1 and Theorem 2. In Theorem 1, ensemble equivalence is proven by assuming the clustering property. While this method has been employed in several previous studies, it is unclear whether adequate comparisons with past research have been made. I do not think that they cite all relevant papers.

Theorem 2, however, has been questioned by previous reviewers for its novelty and validity and is not a particularly intuitive theorem. Transforming with a Haar random unitary U disrupts locality, making it counterintuitive that the resulting state would closely approximate a local Gibbs state. Although I cannot entirely exclude the possibility of misunderstanding, previous reviewer comments suggest that the theorem does not appear to advance new physics. Additionally, the key aspect—that the initial state has small entanglement—remains unclear from reading the paper, as it seems buried within the calculations.

For these reasons, the paper is not particularly readable, and I would recommend a more detailed and accessible revision. Considering the landscape of previous thermalization studies, it is difficult to identify what is genuinely new in this work, making it unsuitable for recommendation for a high-impact journal, such as *Communication Physics*.

Version 1:

Reviewer comments:

Reviewer #5

(Remarks to the Author)

The authors have adequately addressed my comments in this revised version and the accompanying reply. Given the inherent challenges in uncovering entirely novel findings and distinguishing incremental advancements in the study of thermalization, I find the authors' responses to be satisfactory. Therefore, I recommend that this manuscript be accepted for publication in this journal.

Reply to Reviewer #4:

The authors consider the problem of thermalization of systems described by local Hamiltonians. They argue that, since the true Hamiltonian cannot be known exactly, one can only assume a random “prior” distribution of Hamiltonians, preferably on a small set that is centered around the original Hamiltonian.

As their main result, the authors show that a slight randomization of the Hamiltonian can ensure the thermalization of initial states with exponentially decaying correlations.

The small set around the original Hamiltonian is defined via a smoothing parameter δ that limits the perturbation of the elements on the diagonal. The same δ sets the width of the belt around the diagonal within which it is allowed to perturb the nondiagonal elements.

Randomization of the Hamiltonian within these constraints is achieved by Haar-randomizing δ -sized blocks near the diagonal.

The authors show that the δ ensuring thermalization can be chosen to be much smaller than the average energy of the system, so much so that the perturbation barely affects the local states, the energy, and the short-time dynamics.

All previous results in the present context showed thermalization only when the full Hamiltonian is randomized, e.g., by Haar-randomly sampled rotations of the whole basis. Doing so perturbs the Hamiltonian so much that the energy fluctuates macroscopically, and the local states change drastically.

So, such randomization cannot be attributed to mere noise or lack of knowledge.

Thus, this work goes significantly beyond those works by proving thermalization with essentially undetectable randomization of the Hamiltonian that can be attributed to weak noise or lack of knowledge. All results are proved with a good level of mathematical rigor.

Together, the above points check all the boxes of the criteria for publication of Communications Physics.

Therefore, the results in this paper more than merit being published in this journal.

There have already been three pretty thorough reviews of the paper, and the authors addressed most of the issues raised by the previous referees. As such, the revised paper is in good shape as is.

We sincerely thank the referee for their review and the positive assessment of our work, as well as for the recommendation to publish the manuscript in Communication Physics. We are delighted to read this.

Below are some minor issues that the authors should fix before publication.

We are grateful for those additional comments.

- Somewhere below Eq. (19), the authors write “randomizing the eigenbasis locally in energy”, which is a confusing use of language. The term local is already reserved for spatially locality, and the $H \rightarrow UHU^*$ transformation makes the Hamiltonian lose its local structure.

We agree this could cause confusion, so we changed this to “randomizing the eigenbasis by scrambling eigenstates near in energy”.

- Should the ω in the line above Eq. (12) be Ω ?

Here we rely on a theorem in [Anurag Anshu 2016 *New J. Phys.* **18** 083011] requiring Δ to be larger than a specific Hamiltonian dependent constant times the square root of N , $\Omega(\sqrt{N})$ would then not be enough as the constant hidden in the Ω might be too small. Since later we need Δ to scale strictly faster than \sqrt{N} anyway, we assume the same in this theorem for simplicity, even though it would be enough to define a constant C and assume Δ is larger than $C\sqrt{N}$.

- In Eq. (17), the first ρ_∞ has a loose superscript.

We thank the referee for letting us notice this, we have fixed it.

Again, we thank the reviewer for the positive review and the extremely helpful feedback. Having accommodated all comments, we hope that our work is now suitable for publication.

Reply to Reviewer #5:

This paper analytically discusses the thermalization process in time evolution when the initial state has only a small amount of entanglement. There are several unique aspects in the analysis; for example, the use of a generalized micro-canonical ensemble (GmE).

We would like to thank the reviewer for the report.

The primary results are Theorem 1 and Theorem 2. In Theorem 1, ensemble equivalence is proven by assuming the clustering property. While this method has been employed in several previous studies, it is unclear whether adequate comparisons with past research have been made. I do not think that they cite all relevant papers.

The ensemble equivalence between the micro-canonical and canonical ensemble has indeed a long history, and we believe a full literature review on the topic to be outside the scope of our paper. Nevertheless we added a few relevant citations that seemed appropriate. We would like to stress however, that Theorem 1 is not meant as an advancement of previous equivalence of ensembles results between micro-canonical and canonical ensembles, but rather as a stepping stone for Theorem 2, and as we state it is an adaptation of Brandao and Cramer's result to the equilibrium ensemble of the randomized dynamics we consider, which is not the micro-canonical ensemble.

Theorem 2, however, has been questioned by previous reviewers for its novelty and validity and is not a particularly intuitive theorem. Transforming with a Haar random unitary U disrupts locality, making it counterintuitive that the resulting state would closely approximate a local Gibbs state.

We appreciate this point, but would like to politely and respectfully disagree that previous referees questioned the validity of Theorem 2. Some comments regarding its interpretation and the typicality of our ensemble have been raised, which we have addressed in our replies to the best of our knowledge.

It can, indeed, be unintuitive that the scrambling of the Hamiltonian would not disrupt the Gibbs state, at least when local observables are concerned. To make this more intuitive, consider a microcanonical state of H . This state 1. Is locally equivalent to a Gibbs state of H , and 2. is invariant under any rotation inside the microcanonical window. Rotations local in energy cannot then affect the local properties of a Gibbs state, as they leave micro-canonical states invariant, even if they do disrupt the locality of the Hamiltonian. A more direct intuition regarding Theorem 2 is that any state of sufficiently high entropy at the correct energy is locally equivalent to the corresponding Gibbs state - regardless of locality. The rotation increases the entropy with high probability, and it is designed such that it does not disrupt the energy, thus making the final state closer to being micro-canonical, and hence locally equivalent to the Gibbs state.

In order to avoid any misunderstandings, we have added a sentence briefly explaining this intuition in the main text.

Although I cannot entirely exclude the possibility of misunderstanding, previous reviewer comments suggest that the theorem does not appear to advance new physics.

In response to the comments of the previous referees, we already reworked our article, especially in the introduction, to better highlight the novel contributions of our work, we also highlighted the novelty of the results in our reply to the comments the referee mentions. Namely, we give precise lower bounds to the strength of the randomization necessary to guarantee thermalization, and discuss the effects the randomization has on the short time dynamics generated by the Hamiltonian, proving that under certain conditions the effect is negligible. Technically speaking, this relates the Berry-Esseen error to the thermalization properties of the Hamiltonian. We further modified the relevant portions to add some additional references and make this clearer.

Additionally, the key aspect—that the initial state has small entanglement—remains unclear from reading the paper, as it seems buried within the calculations.

We would like to thank the referee for this remark, we added some explanations to make the role of entanglement in the initial state clearer. In short, works on thermalization commonly assume that the state has negligible support outside of a small window. Strong concentration bounds for low entanglement states allow to make this statement precise. In the equivalence of ensembles result for the GmE, in order for all the smaller windows to be assigned the same temperature, they cannot be further away in energy from each other than a certain distance. The known concentration bounds for low entanglement states allow us to cut away the tails of the state beyond this distance and still get (quasi-)polynomially decaying bounds for the distinguishability. We remark that the same method works for any state with bounded energy fluctuations, but then the best bound we can get on the distinguishability with a thermal state decays extremely slowly in the system size, as $1/\log(N)$. For this reason, we have decided to focus on the case of low-entanglement states.

For these reasons, the paper is not particularly readable, and I would recommend a more detailed and accessible revision. Considering the landscape of previous thermalization studies, it is difficult to identify what is genuinely new in this work, making it unsuitable for recommendation for a high-impact journal, such as *Communication Physics*.

We are sorry the referee found the paper hard to read, we sincerely hope our updated version helps clarify the issues mentioned in this report. Again, we thank the reviewer for the report and hope that now our manuscript is suitable for publication.